# Transfer functions linking neural calcium to single voxel functional ultrasound signal

Ali-Kemal Aydin [1,2], William D. Haselden[3], Yannick Goulam Houssen[1], Christophe Pouzat[4], Ravi L. Rungta[1,2], Charlie Demené[5], Mickael Tanter[5], Patrick J. Drew [6], Serge Charpak [1,2,8 ✉] & Davide Boido [1,7,8 ✉]

Functional ultrasound imaging (fUS) is an emerging technique that detects changes of cerebral blood volume triggered by brain activation. Here, we investigate the extent to which fUS faithfully reports local neuronal activation by combining fUS and two-photon microscopy (2PM) in a co-registered single voxel brain volume. Using a machine-learning approach, we compute and validate transfer functions between dendritic calcium signals of specific neurons and vascular signals measured at both microscopic (2PM) and mesoscopic (fUS) levels. We find that transfer functions are robust across a wide range of stimulation paradigms and animals, and reveal a second vascular component of neurovascular coupling upon very strong stimulation. We propose that transfer functions can be considered as reliable quantitative reporters to follow neurovascular coupling dynamics.

[1] INSERM U1128, Laboratory of Neurophysiology and New Microscopy, Université de Paris, Paris, France. [2] INSERM, CNRS, Institut de la Vision, Sorbonne Université, Paris, France. [3] Medical Scientist Training Program and Neuroscience Graduate Program, The Pennsylvania State University, University Park, PA, USA. [4] MAP5, Mathématiques Appliquées Paris 5, CNRS UMR 8145, Paris, France. [5] Physics for Medicine, ESPCI, INSERM, CNRS, PSL Research University, Paris, France. [6] Department of Engineering Science and Mechanics, The Pennsylvania State University, University Park, PA, USA. [7] NeuroSpin, Bât 145, Commissariat à l'Energie Atomique-Saclay Center, 91191 Gif-sur-Yvette, France. [8] These authors contributed equally: Serge Charpak, Davide Boido. ✉email: serge.charpak@inserm.fr; davide.boido@cea.fr

Functional imaging techniques based on blood flow dynamics are extensively used to investigate normal and pathological brain function in humans. However, these techniques do not directly measure neuronal activation, reporting instead functional hyperemia, a delayed increase of blow flow resulting from neurovascular coupling (NVC). The signaling pathways by which NVC occurs are known to involve the cooperation of numerous cell types (neurons, astrocytes, smooth muscle cells, pericytes, endothelial cells) forming the neurovascular unit. However, the precise mechanisms by which these cell types interact and contribute to the coupling are highly debated[1]. In addition, several studies have shown that local activation of neurons generates a back-propagating signal along vessels[2–6], dilating upstream large capillaries and arterioles, and increasing the brain volume irrigated by functional hyperemia[7,8]. This raises the question of how functional imaging techniques based on blood dynamics and used to map brain activity with a mesoscopic spatial resolution report local cellular and vascular responses measured with microscopic resolution, i.e., with two-photon microscopy (2PM).

Functional ultrasound (fUS) has recently emerged as an efficient alternative to BOLD fMRI for functional brain mapping, with millisecond time and mesoscopic ($\sim100 \times 100 \times 200\,\mu m$) spatial resolution[9,10]. It has been used to investigate olfactory[5,11,12], visual[13,14], and somatosensory[9,15,16] responses in anesthetized and awake rodents, and applied to ferrets[17], pigeons[18], monkeys[19] and humans[20,21]. In the mouse olfactory bulb (OB), we have recently shown that global fUS sensory responses are tightly correlated to odor[22]. However, a major question remains: to what extent does the fUS signal report local brain activation? What Doppler signal change (which reflects the increase in cerebral blood volume) can be expected from cellular or vascular responses and reciprocally, what cellular or vascular responses can be deduced from fUS signals? Understanding these relationships is essential to interpreting fUS measurements.

Here, we develop a theoretical approach for transfer function computation and test its validity in establishing TFs between (i) microscopic neuronal and vascular signals and (ii) microscopic neuronal and mesoscopic fUS signals. In the OB of GCaMP6f expressing mice, we first use two-photon imaging to collect neuronal $Ca^{2+}$ signals to odor, simultaneously with RBC velocity changes from capillaries located in the most responsive glomerulus. We then establish the TFs between the two microscopic signals and demonstrate their robustness and limitations. In a second step, we measure the fUS signal from the single fUS voxel containing the responsive glomerulus, i.e., precisely co-registered in the same animal. Finally, we quantify the extent to which mesoscopic TFs (MTFs) can be used to predict neuronal activation.

## Results

### The TF between neuronal and vascular responses within a single glomerulus.
Using two-photon laser scanning microscopy, we imaged neuronal activity and capillary blood flow in the dorsal OB of mice expressing YFP and GCaMP6f under the control of the M72 and Thy1 promoters, respectively. The animals were chronically implanted with a polymethylpentene (PMP) window, allowing 2PM and fUS imaging sessions with reproducible responses over weeks[22]. In these animals, the M72 glomerulus (with terminals converging from olfactory sensory neurons expressing the M72 odorant receptor) was easily distinguishable (Fig. 1a) under a stereomicroscope or using 2PM. Vessels were labeled with Texas Red and linescan acquisitions were used to simultaneously monitor capillary RBC velocity and neuronal $Ca^{2+}$ upon odor stimulation. While stimulation with 1–6% ethyl tiglate (ET) during 5 s activated several glomeruli in the close

vicinity of the M72 glomerulus, lowering the odor concentration in the range of 0.1% enabled us to isolate the most sensitive glomerulus, which was then selected for the rest of the study. Figure 1b illustrates a typical neurovascular response to the stimulation protocol (1% ET, 5 s) used to build a database from 15 mice, and compute a microscopic transfer function ($\mu$TF) between neuronal $Ca^{2+}$ and capillary vascular responses. Deconvolution approaches based on Fourier transformation or Toeplitz matrix were too sensitive to the noise of $Ca^{2+}$ and RBC velocity traces to compute a reproducible $\mu$TF. We thus chose to optimize a transfer function based on the gamma-distribution function, which is commonly used as a basis for the hemodynamic response function (HRF) in BOLD fMRI. We did not need a second, negative gamma-component as there was no post-stimulus undershoot. We also added a time-shift parameter ($p_3$), to better match the data (accounting for a vascular delay $\geq 1$ ms, H($t$) is the heaviside unit step function).

$$\mathrm{TF}(t) = \mathrm{H}(t - p_3)p_4\left(\frac{(t - p_3)^{p_1 - 1}p_2^{p_1}e^{-p_2(t - p_3)}}{\Gamma(p_1)}\right).$$

The $\mu$TF optimization was performed using a nondeterministic, machine-learning approach based on the simulated annealing algorithm (see Supplementary Fig. 1 and Methods for details). One $\mu$TF was optimized for each mouse, under the same stimulation protocol (1% ET, 5 s) (Fig. 1c, top). Using this approach, vascular response predictions were excellent (Fig. 1c, bottom, Pearson coefficient: $0.93 \pm 0.03$, mean $\pm$ SD). They remained robust using either the $\mu$TF optimized with each mouse own $Ca^{2+}$ and RBC velocity data (self-validation), or the $\mu$TF optimized with data from other mice (leave-one-out cross-validation) (Fig. 1d). One $\mu$TF, peaking at 0.9 s and providing the best robustness (self and cross performance, for more details, see Methods), was then selected as the standard $\mu$TF to predict vascular responses from $Ca^{2+}$ responses (Fig. 1e). The robustness of this $\mu$TF suggests that the ensemble of cellular mechanisms underlying NVC is tightly and similarly controlled across animals. However, whether this holds true across different stimulation durations or intensities remains unknown.

### Testing TF robustness through various stimulation conditions.
We first decreased the odor delivery from 5 s, the duration at which the standard $\mu$TF was initially optimized, to 2 s, 1 s, and 120 ms, the latter duration corresponding to a single sniff stimulation. For such brief stimulation, which suffices for perception and odor discrimination[23–25], the olfactometer was locked to the respiration by means of a thermocouple placed close to the mouse nostril. Figure 2a shows that $Ca^{2+}$ and RBC velocity responses increased according to the odor duration. Vascular responses predictions remained robust with all durations (Pearson coefficients for 120 ms, 1 s, 2 s, 5 s, respectively: $0.74 \pm 0.15$, $0.79 \pm 0.16$, $0.89 \pm 0.07$ to $0.89 \pm 0.08$, mean $\pm$ SD, $n = 5$ mice). This indicates that microscopic functional hyperemia is a robust reporter of neuronal activation duration. As stimulus intensity was the second stimulation parameter across which the $\mu$TF needed to be validated, we compared $Ca^{2+}$ and RBC velocity responses at 1% ET (the concentration used to compute the standard $\mu$TF) and 6% ET (a strong concentration) in five other mice. As expected, at 1% ET the change in RBC velocity prediction was excellent. However, at 6% ET a secondary delayed phase of the RBC velocity response appeared and was poorly predicted (Fig. 2b). This secondary phase, peaking around 30 s is reminiscent of the delayed and astrocyte-mediated BOLD fMRI signal reported under strong stimulation in the neocortex[26]. It could be isolated by subtracting prediction values from the

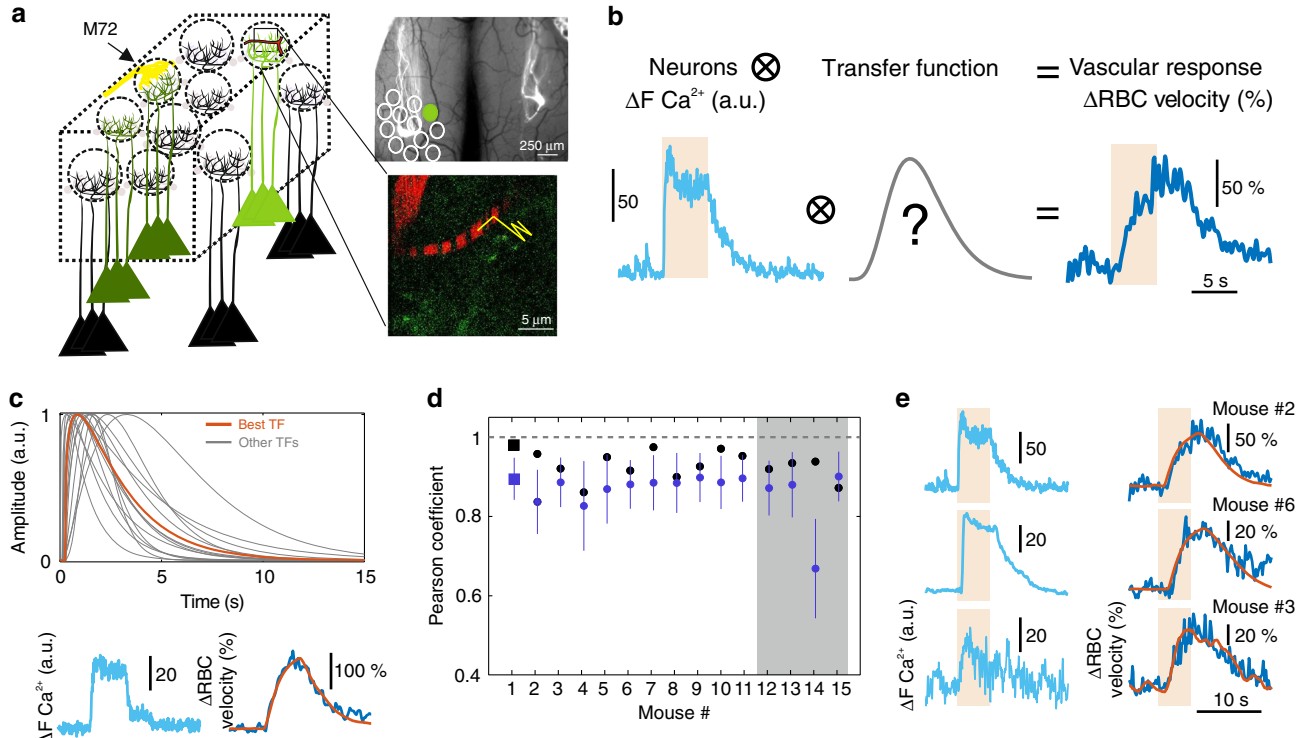

**Fig. 1 The transfer function between neuronal and vascular responses within a single glomerulus: computation and robustness across mice. a** Left, schematics of the dorsal olfactory bulb (OB) of a mouse expressing YFP and GCaMP6f under the control of M72 and Thy1 promoters, respectively. Depending on the odor concentration, ethyl tiglate (ET) activates several glomeruli or only the most responsive one, which is in close vicinity to the M72 glomerulus. Right top, axons converging in the two M72 glomeruli are imaged with a stereoscope through a chronically implanted PMP window (photographs were taken for all mice). Right bottom, an image showing a capillary labeled with Texas red. A broken line is used in the linescan acquisition mode to monitor $Ca^{2+}$ in the neuropil (dendritic GCaMP6f) and RBC velocity in the capillary. **b** For each mouse, a microscopic transfer function (µTF) is convolved with $Ca^{2+}$ signals, the µTF being optimized to predict RBC velocity changes in response to odor 1% ET (5 s). **c** Top, µTFs optimized for each mouse ($n = 15$). The orange curve is the µTF optimized from $Ca^{2+}$ and RBC velocity responses illustrated in bottom, which gives the prediction curve overlaid in orange on the RBC response. **d** Quantification (mean ± SD, $n = 15$ mice) of prediction robustness for each µTFs optimized using either the TF derived from the same mouse (black symbols, single self prediction) or data from other mice (blue symbols, mean cross-validation). The µTF from mouse #1 (square symbols) gives the best 'self' vascular prediction (see **c**) and good predictions across mice. It was selected as the standard µTF to predict vascular responses from $Ca^{2+}$ responses. Gray shadow (#12–15) for Pearson coefficients obtained with data acquired in mice from Boido et al, 2019[22]. **e** Examples of vascular response predictions from three mice using the standard microscopic TF and optimized for the amplitude (see Methods). Source data are provided as a Source Data file.

experimental data (Fig. 2c, black trace) and was observed in all mice, starting at ~15 s and peaking at ~25–30 s (gray traces). To improve the standard µTF prediction and account for this secondary vascular phase, we first optimized a new set of µTFs for each mouse, linking $Ca^{2+}$ responses and the subtracted secondary vascular component (Fig. 2d, left). These secondary µTFs displayed a strong peak jitter and shape heterogeneity. They could be combined with the standard µTF, taking into consideration the responses' amplitude ratio, to predict the vascular response over its full duration (Fig. 2d, inset). We then investigated whether the standard µTF could predict responses elicited by a second odor, isoamyl acetate (1%, 5 s). This odor very poorly activates the glomerulus most sensitive to ET. Supplementary Figure 2 shows that despite the very modest $Ca^{2+}$ and RBC velocity responses, the µTF vascular response prediction remained robust ($n = 4$ mice). Finally, we challenged the µTF robustness in predicting vascular responses with our data collected and used in previous publications: (1) In similarly sedated mice[22], the prediction quality of the vascular response initial component remained robust at increasing odor concentrations of 0.4%, 1%, and 6% ET (Pearson coefficient: 0.83 ± 0.08, 0.89 ± 0.06, 0.89 ± 0.07, respectively, mean ± SD, $n = 6$ mice); (2) In awake trained mice[5], although vascular responses were briefer and occasionally

showing a delayed undershoot (Supplementary Fig. 4a), a phenomenon never observed during anesthesia, vascular responses were predicted with Pearson coefficients of 0.91, 0.84, 0.77, 0.92 ($n = 4$ OBs, 3 mice). Altogether, these results indicate that the µTF can be efficiently used to predict vascular responses from neuronal $Ca^{2+}$ signals, across mice, stimulation parameters and brain states. We next asked whether this quantitative approach of linking neuronal and local vascular responses with the µTF could be extended to mesoscopic imaging techniques. To establish MTFs, we measured fUS CBV responses from the single voxel (~$100 \times 100 \times 200$ µm) comprising the most responsive glomerulus imaged with 2PM.

**Single voxel fUS responses faithfully report microscopic responses.** To co-register the 2PM and fUS imaging systems, the ultrasonic probe was attached to the microscope objective by means of a 3D-printed holding piece. A 50-µm glass bead, embedded in agar, was first localized with 2P imaging at given coordinates in 3D. The ultrasonic probe was then precisely translated over the bead and moved back and forth in the x, y, and z directions (Fig. 3a). This allowed to collect the Gaussian fUS intensity profiles in 3D for a selected voxel, which center was then fixed at the profile maxima, allowing the back and forth

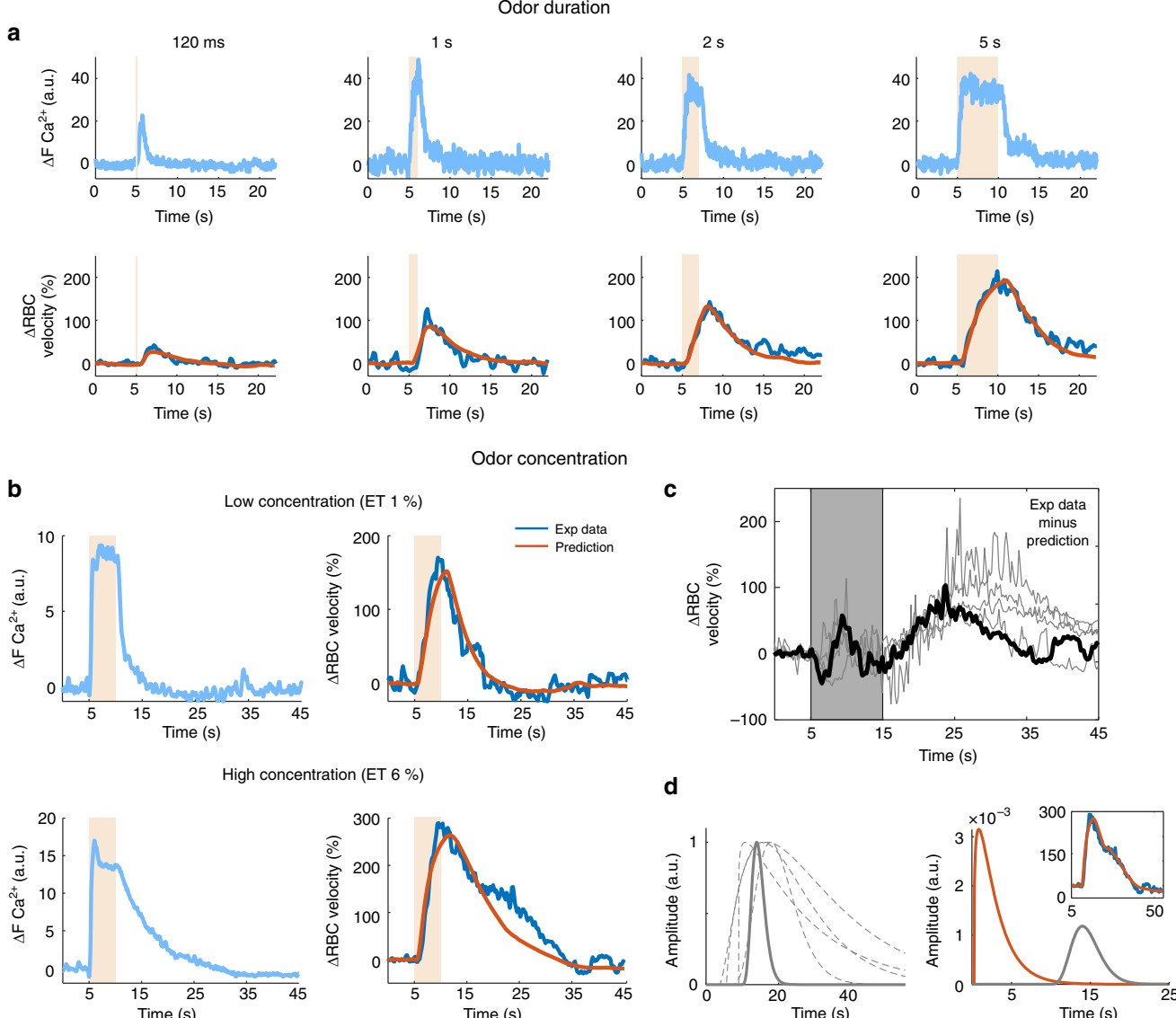

**Fig. 2 Robustness of TFs with respect to stimulation duration and intensity. a** $Ca^{2+}$ (top) and RBC velocity (bottom) responses to single sniff (120 ms), 1 s, 2 s, and 5 s odor stimulations. Using the standard μTF (optimized with ET 1%, 5 s), vascular response predictions (orange traces) are robust for all durations. **b** Increasing the odor concentration from 1% (top traces, average of two trials) to 6% ET (bottom traces, average of three trials) reveals a delayed secondary vascular response, which is not correctly predicted (orange trace) with the standard TF (1% ET, 5 s). **c** Vascular responses after subtraction of their prediction using the standard TF ($n = 5$ mice, data from **b** in black). Note that trace fluctuations between 5 and 15 s (gray background) result from slight differences of onset, slope, and response peak between real and predicted responses. The second vascular component is clearly delayed by ~10 s. **d** Left, novel TFs optimized on vascular responses after subtraction (the thick trace is the TF from the data in **b** and **c**). Note the heterogeneity in peak jitter and shape. Right, combination of the two μTFs (standard μTF in orange, second component μTF in gray) can predict correctly the entire vascular response (see inset, same trace as in **b**) after adjustment of their corresponding amplitude.

displacement of the two imaging systems at the same co-registered location with a micrometric resolution (Fig. 3b), as well as the collection of micro/mesoscopic functional dataset. We first acquired 2PM neuronal and vascular data from the most responsive glomerulus upon 1% ET (5 s). fUS power Doppler signals were then collected from (1) the single voxel centered on the most responsive glomerulus, and (2) from a small voxel group (the specific voxel plus its five nearest neighbors) to account for the vascular backpropagation (Fig. 3c, d). Figure 4a shows that the time course of 2PM $Ca^{2+}$, RBC velocity and mesoscopic fUS ΔPD/PD responses acquired at two odor concentrations (1 and 6% ET) and two stimulation durations (120 ms and 5 s). The responses increased with either odor duration or concentration ($n = 5$ mice). Vascular responses (microscopic and mesoscopic)

were remarkably similar (Pearson coefficients between RBC velocity and fUS data (single voxel): $0.52 \pm .21$ (1%, 120 ms ET), $0.73 \pm 0.05$ (6%, 120 ms ET), and $0.67 \pm 0.13$ (1%, 5 s ET), $0.70 \pm 0.05$ (6%, 5 s ET) mean ± SD). Using our previous TF optimization approach (single gamma-component), we obtained an MTF between neuronal $Ca^{2+}$ and the co-registered single fUS voxel signal. In comparison with the μTF, this neuron-derived MTF had a slower decay (Fig. 4b). The prediction quality of single voxel fUS responses were low for brief stimulations, in particular at 1% ET (with the μTF: $0.33 \pm 0.13$; with the MTF: $0.35 \pm 0.16$, mean ± SD) (Fig. 4c). However, all predictions improved when considering fUS responses from the six voxels, i.e., with a better signal-to noise ratio and a dynamic similar to that of the co-registered voxel. Note that upon high odor concentration (6% ET,

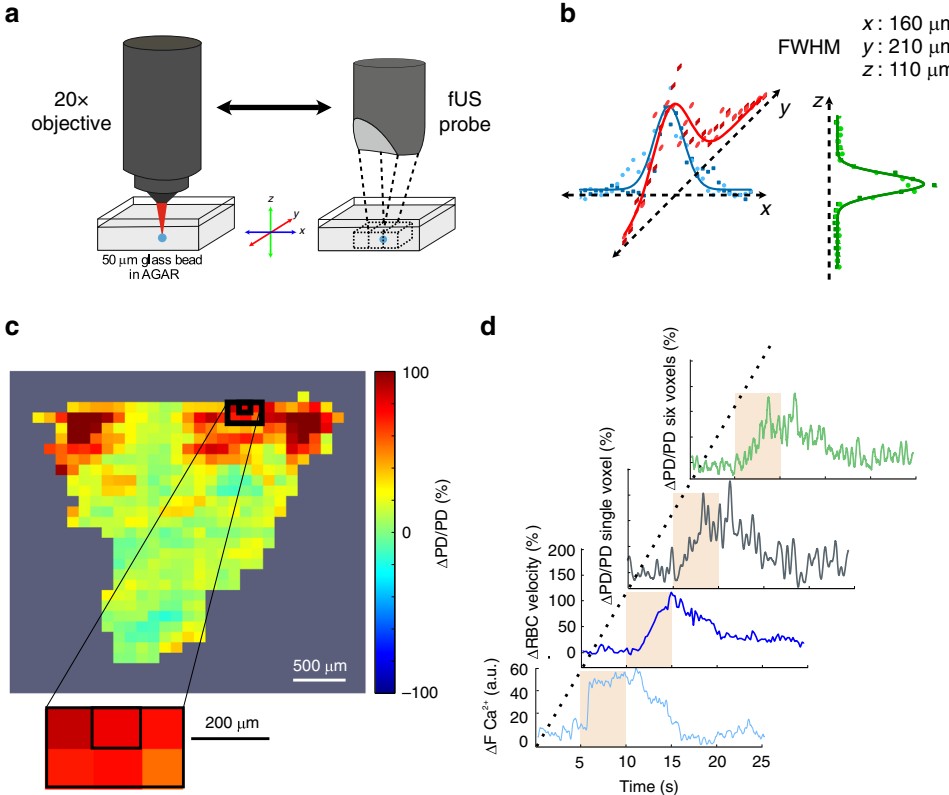

**Fig. 3 Co-registration of fUS (single voxel) and two-photon imaging. a** Schematics of the two imaging systems. The ultrasonic probe is attached to the ×20 microscope objective by means of a 3D-printed holding system. A 50-μm glass bead, embedded in agar, is first localized with 2P imaging. The ultrasonic probe is then translated over the bead and placed at a position where the fUS signal maxima in x, y, and z are centered in a given voxel. The fUS and 2P imaging systems can then be displaced back and forth to the same co-registered location with a micrometric resolution. **b** Intensity profiles of fUS signals in x, y, z. FWHM for full width at half maxima. Square and circle points were acquired during back and forth acquisitions. **c** ΔPD/PD fUS activation map of an olfactory bulb coronal section in response to 1%, 5 s ET. The enlarged area shows the voxel centered on the most responsive glomerulus (first imaged with 2P) plus its five neighboring voxels. **d** From bottom to top, Ca²⁺ and RBC velocity glomerular responses (2P imaging), ΔPD/PD fUS responses from the co-registered voxel and the six voxels.

5 s), fUS responses showed a delayed secondary component (Supplementary Fig. 3), similar to RBC velocity responses at this concentration (Fig. 2b, c). Can the MTF be used with techniques reporting other types mesoscopic signal? We analyzed the freely available data from Winder et al.[27], collected in the somatosensory cortex of awake mice, and in which neuronal activity was monitored with electrophysiology and mesoscopic CBV responses with intrinsic optical recordings (IOS). Whisker stimulations evoked brief changes in the local field potential (LFP) gamma-band power and in light reflectance. Even though neuronal (Ca²⁺ versus gamma-band power in the LFP) and vascular (RBC velocity versus IOS reflectance) are different proxies signals, the MTF was still robust in predicting mesoscopic CBV changes (Supplementary Fig. 4b, d). Nonetheless, an ad hoc TF could be further optimized using our computing approach (Supplementary Fig. 4b–e, brown versus orange predictions), yielding a significant improvement to the prediction quality. In total, the MTF computed in the OB remains surprisingly robust in predicting vascular responses in the neocortex, even though it improves with optimization based on the dataset.

**Statistical analysis of single versus six voxels fUS responses.** We have recently shown that due to the poor specificity of odorant receptors and the vascular backpropagation, ET 1 and 6% activate numerous voxels in the OB[22]. The co-registration enabled us to test whether at such concentrations, blood flow responses and statistical analysis enable to extract the voxel containing the most sensible glomerulus from its five neighboring voxels. We first quantitatively compared fUS responses (areas under curves of ΔPD/PD responses) in the six voxels at 1% ET (5 s) (Fig. 5a). The fUS response from the voxel containing the most responsive glomerulus was the largest in only one out of five mice. Statistical analysis, using statistical parametric mapping, the general linear model, RBC velocity responses as regressors and a statistical threshold ($p < 0.01 ±$ FWE), revealed that, as for the fUS signal (AUC), the $t$ value of the voxel containing the most sensitive glomerulus was not systematically the highest (Fig. 5b). These findings remained valid at all four stimulation paradigms (Fig. 5c, d). Altogether these results point to the fact that at 1 and 6% ET, activation maps cannot sort out the voxel containing the most sensible glomerulus.

## Discussion

The OB glomerulus is an efficient biological model to compute TFs between neuronal activation and microscopic/mesoscopic vascular responses and test their use as surrogates of NVC, providing that the model strengths and weaknesses are recognized: as the first olfaction relay, odor processing by centrifugal inputs is moderate in contrast to sensory cortex models, and odor triggers local blood flow responses that are odorant-specific, concentration-dependent and correlated to local presynaptic and postsynaptic responses[7,28–30]. However, odorant receptor specificity is odor concentration-dependent and additionally, vascular responses are not entirely specific as synaptic activation triggers a

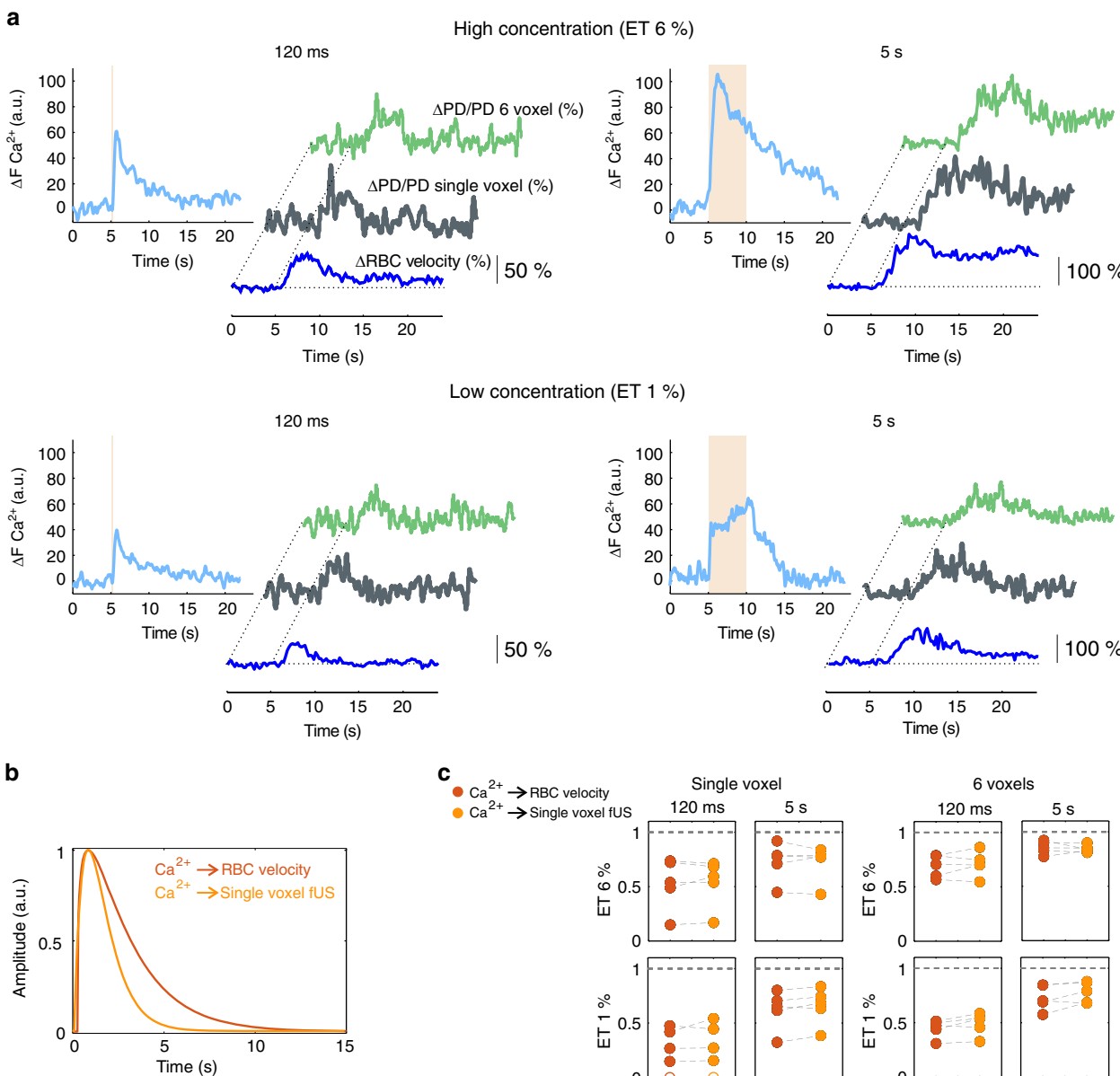

**Fig. 4 Co-registration of microscopic and mesoscopic vascular responses to odor. a** Microscopic responses were acquired in a glomerulus located in the center of a fUS voxel. The four panels illustrate $Ca^{2+}$, RBC velocity, and the three types of fUS responses (from the specific co-registered single voxel and the group of six voxels) to four odor stimulation protocols (120 ms and 5 s stimulations at two concentrations). All responses increased with time and odor concentration. The ΔPD/PD fUS traces from single voxels are in bold. **b** Comparison of the µTF and the mesoscopic TF (ᴍTF) optimized between $\Delta Ca^{2+}$ (ET 1%, 5 s) and the co-registered single fUS voxel signal. **c** Prediction quality (Pearson coefficients) of single and six voxels fUS responses using either the µTF (red), the neuron-derived ᴍTF (orange) ($n = 5$ mice). Note that in one mouse (bottom left plot), the single voxel signal ∼0 (open circles). Source data are provided as a Source Data file.

vascular signal that propagates retrogradely along the vascular arbour, causing a dilation of upstream arterioles[5]. Consequently, even when targeting the glomerulus most sensitive to a given odorant, in our case ET, odor stimulation will always cause an increase of blood flow over a brain volume larger than that of activated neurons. The use of TFs is thus limited by the spatial and temporal resolution of NVC. Ideally, modeling of NVC should give a better read out to predict functional hyperemia, as well as allowing better inference about neuronal activation from functional hyperemia. Unfortunately, there is a general agreement that transmitter release triggers a complex feed-forward cascade, and the differential weight of all the cellular partners involved in NVC remains unclear. The timing of $Ca^{2+}$ signaling in astrocyte processes, endothelial cells, or contractile cells (enwrapping

pericytes and smooth muscle cells) is compatible with a role of these cells in NVC but without testing the functional consequences of silencing specific mechanisms, NVC modeling remains premature. By bypassing the NVC cellular cascade, TFs thus appear as efficient surrogates linking local neuronal activation to vascular responses.

TFs have been used to predict CBV and BOLD responses to spontaneous and evoked neuronal activation in numerous studies[27,31–37]. The novelty of our approach relies on several technical advances: (i) we measured neuronal responses from a given compartment (the apical dendritic tuft) of specific cells (OB principal cells); (ii) we characterized the µTF at the site of synaptic activation; (iii) we used a machine-learning non-deterministic approach, based on the simulated annealing

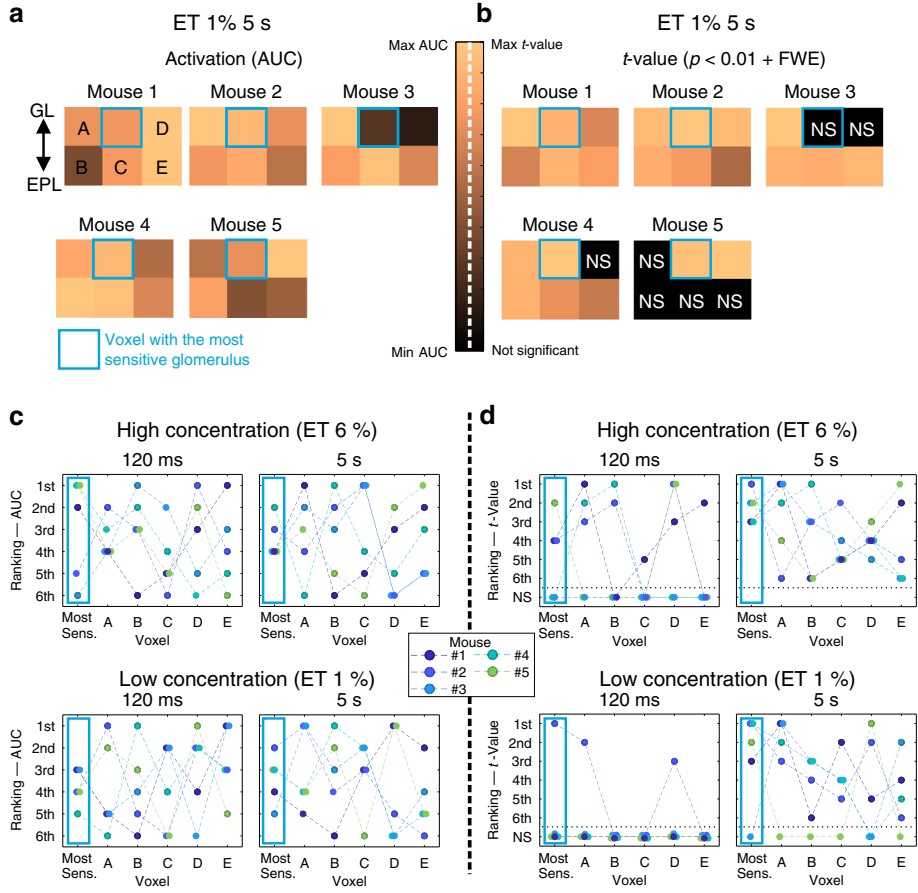

**Fig. 5 Spatial resolution of fUS responses. a** Quantification of fUS responses (area under the curve, ET 1%, 5 s) in the voxel containing the most sensitive glomerulus (outlined with a blue frame) and the five neighboring voxels (A–E, n = 5 mice, GL glomerular layer, EPL external plexiform layer). The color map shows that only in one case, the largest response was located in the blue voxel. The color bar was scaled with respect to the highest AUC value across the six voxels. **b** Statistical analysis (GLM) of fUS responses in the corresponding voxels using the RBC velocity response (CBV flowing at 0.5–1.5 mm s$^{-1}$) as a regressor. The t value for the voxel containing the most sensitive glomerulus was not systematically the highest one. The applied statistical threshold was $p < 0.01$ + FWE correction. NS for not significant. **c** Quantification of fUS responses (AUC) for four stimulation conditions. For the five mice, the voxel containing the most sensitive glomerulus and the other five voxels were ranked with the 1st place assigned to the voxel with the highest AUC value. **d** Statistical quantification of fUS responses with ranking based on t values. The 1st place was assigned to the highest t value. Source data are provided as a Source Data file.

algorithm, to optimize our TFs; (iv) we validated the robustness and reliability of the μTF across mice and stimulation parameters; (v) we co-registered micro and mesoscopic vascular signals with the best spatial and temporal resolution so far and could compare fUS responses of the voxel comprising the activated synapses to neighboring voxel responses. Note that the comparisons were specifically limited to CBV changes in vessels with slow velocities, i.e., in capillaries[16,22,38]. These advances allowed us to uncover several aspects of NVC. The μTF had fast dynamics compatible with the previous finding[5] that Ca$^{2+}$ drops within ~300 ms in mural cells along the upstream arteriolar compartment, triggering vessel dilations. The μTF was robust across mice and a wide range of stimulation duration and intensities. It is important to stress that between different animals, the μTF robustness concerned the dynamics of NVC but not its amplitude. Interestingly, upon very strong stimulation, the μTF and its vascular response prediction revealed a secondary and delayed CBV response. Note that ET at high odor concentration did not cause any respiration change, suggesting that there was no irritation from the stimulus. Bimodal vascular responses have previously been reported[39,40], Schulz et al.[26] suggesting that it results from the activation of astrocytes. Whatever mechanisms are responsible of the deviation from the standard μTF, our observations imply that NVC, even in a small

and well characterized neuronal network, is robust, but only within a given activation range, since responses become bimodal with strong odor stimulation. This observation is essential to correctly process brain activation maps with blood-flow-based imaging techniques and interpret the linearity of signal processing upon strong stimulation.

In the second part of our study, we investigated the extent to which functional hyperemia measured at the site of synaptic activation is translated at the mesoscopic level, using CBV measurements with functional ultrasound imaging. fUS is now used to image functional neural circuits in the entire brain, yet, the biological nature of the fUS signal needs further characterization with simultaneous recording/imaging of neuronal activity. We find that fUS responses within the single voxel comprising the most sensitive glomerulus share the general shape and timing of RBC velocity responses. Note that they were so similar in time that they could be linked by a simple Dirac Delta function. As the μTF tightly couples neural and RBC responses and strongly resembles the single voxel neuron-derived MTF, the fUS signal can be considered as a good reporter of local neuronal activity.

Both μTF and MTF were obtained from data collected in sedated mice. Surprisingly, they still predicted correctly vascular response dynamics in the OB and the neocortex of awake

animals, even when collecting different types of neuronal and vascular markers. In awake animals, vascular responses are briefer, and in some cases, followed by a delayed negativity/undershoot of unknown cellular mechanism and which was never observed in anesthetized mice. As a result, predictions could be improved with further optimization using the same computing approach.

At 1 and 6%, ET strongly activated the OB, and due to vascular backpropagation[4,5] and nonselective activation of several glomeruli, generated a widespread fUS signal in the entire coronal section (see Fig. 3c). This is in line with what we have previously reported (see Fig. 5 in Boido et al.[22]). Consequently, the fUS response from the voxel containing the most sensitive glomerulus could not be discriminated with respect to the five surrounding voxels, whether assessed on the Power Doppler signal (AUC) or its correlation with RBC velocity responses (GLM analysis). This biological limit points to the risk of inferring with certainty the location of neuronal activation from single fUS voxels, in the OB.

Overall, TFs are efficient tools to report NVC, their robustness revealing a quantitative link between postsynaptic $Ca^{2+}$ signal dynamics and both local microscopic vascular responses and mesoscopic CBV fUS responses.

## Methods

**Animal preparation.** All animal care and experimentations were performed in accordance with the INSERM Animal Care and Use Committee guidelines (protocol numbers CEEA34.SC.122.12 and CEEA34.SC.123.12). Adult mice ($n = 15$, 3–12 months old, 20–35 g, both males and females, housed in 12 h light–dark cycle at 24 °C and 50% humidity, fed ad libitum) were used in this study. Thy1-GCaMP6f (GP5.11)[41] mice were obtained from Jackson laboratory and crossed with M72->S50-IRES-hChRVenus mice[42] kindly provided by Thomas Bozza (Department of Neurobiology, Northwestern University, USA). Surgery, anesthesia and the experimental sedation protocol were performed as previously[22]. For craniotomies, mice were initially anesthetized with an intraperitoneal bolus of ketamine–medetomidine (100 and 10 mg kg$^{-1}$ body mass, respectively). Anesthesia lasted about 2 h, enough to perform the whole surgery. During surgery, mice freely breathed air supplemented with oxygen (final concentration of 40%) and body temperature was maintained at 36 ± 0.5 °C using a rectal probe and a heating pad. The bone over the OBs was carefully drilled while taking care of not applying pressure and avoiding heat with repetitive applications of a cool ACSF solution[43]. A sheet of PMP, either 80 or 125 μm thick, was precisely cut to fit the cranium opening and sealed in place with Tetric Evoflow dental cement (Ivoclar Vivadent AG, Schaan, Liechtenstein). The same dental cement was also used to form a head-cap in which a titanium head-bar was embedded after the bone was treated with a primer, Optibond FL (Kerr Italia S.RL., Salerno, Italia). Antibiotics and anti-inflammatory were administered as in[43] using Baytril 10% (Bayer, Germany) and Dexazone (Virbac, France) respectively. Buprecare (Axience, France) was used to relieve pain. Mice were permitted to recover for at least 2 weeks before the experimental sessions began.

Experiments were performed under light sedation: mice were anesthetized for 2 min with 3% isofluorane and then head-fixed while isofluorane was lowered to 2%. A subcutaneous catheter was placed dorsally and medetomidine injected first as a bolus (0.05 mg kg$^{-1}$) and then immediately at a rate of 0.15 mg kg$^{-1}$ h$^{-1}$, which was maintained during the whole experiment. Isofluorane was progressively removed within 40 min. An additional 20 min delay was kept prior to recording. Note that sedation was stable and reliable after 2–3 priming sessions during which the exact same protocol was applied but without recording. During the experiments, breathing was monitored with an 80-μm-tip thermocouple (Omega 5TC-TT-KI-40-1M) connected to a N9212 amplifier (National Instruments, USA) and digitized with a cDAQ-9171 (National Instruments, USA). The thermocouple was placed in front of one of the mouse nostrils and the body temperature was maintained at 36.5 ± 0.5 °C using a rectal probe + heating pad (FHC, Bowdoin, ME). Mice were supplemented with 40% $O_2$ throughout all experiments.

**Odor stimulation.** All the experiments were conducted following the same stimulation protocol and materials, so no blinding was performed. Odors were delivered through a home built olfactometer. Odor and exhaust lines were equilibrated for pressure at the start of each experiment and the odor concentration and temporal profile was calibrated at the tubing end before every experiment, using a photo-ionization detector (miniPID 200B, Aurora Scientific, Aurora, Canada). The concentration reached 90% of steady state within ~50 ms and returned to baseline within the same range. The final odor concentration values were calculated after considering the dilution from the supplemental $O_2$ line that did not pass through the olfactometer. To compare the responses to different odor durations or concentrations, the stimulations were randomly interleaved, discarding anesthesia-

related changes. All microscopic (2PM) and mesoscopic (fUS) acquisitions lasted 30 s, with the exception of strong stimulations (6% ET, 5 s), for which 60 s acquisitions were used to allow complete recovery to baseline. All the traces reported in the figures represent the average of at least three odor applications, with the exception of Fig. 3a, top panel (two traces). Single sniff (120 ms) were usually averaged over 8–12 applications because of the low signal-to-noise ratio, in particular at 1% ET. The odor delivery was finely time-locked with the inhalation period, detected with the nasal thermocouple.

**fUS data acquisition and post processing.** fUS imaging was performed as follows[22]. In brief, a linear ultrasound probe (128 elements, 15 MHz central frequency, Vermon, Tours, France), connected to the ultrasound scanner (AixplorerTM, Supersonic Imagine, Aix-en-Provence, France), was placed 3 mm above the window. Custom transmit/receive ultrasound sequences were written in Matlab (Mathworks, USA). The backscattered echoes of ultrasound plane waves were collected and beam formed to produce OB echographic images, in the coronal plane. To increase the SNR of each echographic image taken at 500 Hz, the echographic images were compounded by transmitting several tilted plane waves and adding their backscattered echoes. The compounded sequence resulted in enhanced echographic images, thereby increasing the sensitivity of the Doppler measurement without aliasing in the mouse brain. In this study, the ultrasound sequence consisted of transmitting eleven different tilted plane waves (from −10° to 10° in 2° increments) with a 5500 Hz pulse repetition frequency (500 Hz final frame rate). Tissue signals were removed from backscattered waves using singular value decomposition and elimination of the largest eigenvalues. The Power Doppler was further filtered with a Butterworth filter (fifth order, 10–30 Hz band-pass). Each voxel signal was obtained by the incoherent temporal average of the blood signal. Voxel size at the focal plane was: 100 × 110 μm ($x$ and $z$ direction) and 200 μm ($y$ direction, i.e., slice thickness).

**fUS analysis.** All analyses were performed with custom made software developed in Matlab 2018a (Mathworks, USA). Beam-formed data resulted in 500 Hz frame rate time series. GLM analysis of fUS data was performed with SPM12. fUS recordings were single voxel under-sampled at 20 Hz using a cubic interpolation ('interp1' function, 'pchip' option, Matlab). Each frame was then converted to a NIfTI file. The absence of movement was previously assessed using a custom made Matlab script. No realignment or spatial smoothing was performed with SPM. Regressors used in GLM analyses were the RBC velocities recorded in the same mice. We applied a statistical threshold of $p < 0.01$ + FEW correction, in response to each stimulation condition.

**2PM acquisition and data analysis.** 2PM imaging was performed using an ultra-flexible microscope[44] and data were collected and analyzed using a custom software. Eighty femtoseconds laser pulses were delivered by a Ti:Sapphire laser at 80 MHz (MaiTai HP DeepSee, MKS-Spectra Physics, Santa Clara, California). An acousto optic modulator (MT110B50-A1.5-IR-Hk, AA Optoelectronic, Orsay, France) was used to modulate the laser power. Laser pulses were scanned on the sample with galvanometric mirrors (6215H Cambridge Technology, Bedford, Massachusetts). The excitation light was forwarded through a dichroic mirror (cut-off wavelength, 775 nm FF775-DiO1 Semrock) and focused on the sample with a 20 × 1.0 NA objective (XLUMPLFLN20XW Olympus, Tokyo, Japan). GCaMP6f and Texas Red were excited at 920 nm. The collected emission was separated in two channels with a dichroic mirror (cut-off wavelength, 560 nm FF560-DiO1 Semrock). The signal was filtered with a 525 nm band pass filter (FF03-525/50 Semrock) in the « green channel » and a 620 nm band pass filter (620 nm FF01-624/40 Semrock) in « the red channel ». In both channels, GaAsP PMTs (H10770PA-40; Hamamatsu Photonics, Japan) were used to collect light, and laser reflections were blocked with short pass filters (FF01-750/SP-25 Semrock). PMT signals were amplified, integrated with a custom-built electronic circuit and sampled with a National Instruments acquisition card (PCI 6115). Vessels were labeled with Texas Red dextran (70 kDa, Molecular Probes, ThermoFisher, Waltham, Massachusetts), administered intravenously by retro-orbital injections. Following mapping of the ET activated region in frame scan mode and selection of the most sensitive glomerulus to the odor, broken linescan recordings were performed to record both RBC velocity and $Ca^{2+}$ signals in the neuropil, as previously described[7]. Linescan acquisitions (1.5–3 ms per line) were repeated during the different stimulation protocols, with 3 min between each trial. $Ca^{2+}$ data were interpolated (10 ms cubic interpolation, 'interp1' function, 'pchip' option, Matlab), turned into $\Delta Ca^{2+}$ traces by subtracting the average value of the last 5 s of baseline. RBC velocities were extracted and analyzed by a custom software[22]. RBC velocity data were interpolated (200 ms, cubic interpolation, 'interp1' function, 'pchip' option, Matlab), turned into ΔRBC/RBC traces by subtracting the average value of the last 5 s of baseline and then dividing by it. A median filter of three time points was applied to some traces to attenuate overestimated RBC velocity values.

**Co-registration of fUS and two-photon imaging.** To ensure that we image the same brain volume with fUS and 2PM, we attached the ultrasonic probe to microscope objective with a custom printed ABS holder (Makerbot 2×, USA). A 50-μm glass bead (Marteau & Lemarié, Sorbiers, France), embedded in 2.5% agar,

## Table 1 Parameters of the different TFs computed in this study.

| From > To | $p_1$ | $p_2$ | $p_3$ | $p_4$ (amplitude) |
|---|---|---|---|---|
| Single-gamma HRF | 6 | 1 | 0 | 1 |
| Ca$^{2+}$ to RBC velocity (ET 1%, 5 s) | 1.3 | 0.5 | 0.27 | 0.19 |
| Ca$^{2+}$ to Single voxel fUS* (ET 1%, 5 s) | 1.99 | 1.27 | 0.11 | 0.045 |

Source data are provided as a Source Data file.
*0 < Time shift (P3) < 0.5.

was first localized with 2PM and then detected with the ultrasonic probe, translated to a position such that the bead fUS signal maxima in $x$, $y$, and z were centered in a given voxel (Fig. 4a). The fUS and 2PM imaging systems could then be displaced back and forth to the same co-registered location with a micrometric resolution. Note that in our experimental conditions, two factors may have influenced the speed of sound and thus cause approximation along the z axis: the temperature of the media (brain tissue) and the thickness of the PMP cranial window, both of them creating a $\Delta z$ approximation $\leq 25$ μm, for a mean speed of sound of ~1500 m s$^{-1}$.

**TF computation**. The transfer function captures the dynamic link between neuronal responses and micro/mesoscopic vascular responses. Using Ca$^{2+}$ and RBC velocity data collected with 2PM and fUS CBV data within the co-registered brain volume, we built TFs across the different imaging modalities. We used a fitting approach with a single-gamma distribution function, as there was no undershoot in flow in our data. We added a time-shift parameter with the heaviside unit step function, resulting in a total of four parameters for the optimization problem. These parameters capture the slope, the decay, the amplitude and time-shift, respectively, but they are not independent from each other: the optimization problem presents multiple minima configurations, a condition that is not favorable for quasi-Newtonian optimization algorithms.

All analyses were performed with a custom-written software in Matlab 2018a.

The four-parameters function, $\mathrm{TF}(t) = \mathrm{H}(t - p_3) p_4 \left( \frac{(t-p_3)^{p_1-1} p_2^{p_1} e^{-p_2(t-p_3)}}{\Gamma(p_1)} \right)$, was optimized by minimizing the sum of the square residuals, using simulated annealing, a machine-learning algorithm ('simulannealbnd' function, Matlab). Note that this is a nondeterministic algorithm: the same set of initial values can bring to different sets of optimized values for the parameters, unless the initial set represent a stable minimum for the optimization function. Note that it was not necessary to modify the Matlab default initial temperature (100) and temperature function ('temperatureexp' function, Matlab). Lower and upper boundaries were set to 10$^{-3}$ and 10, respectively, for all parameters. Upper bounds were occasionally increased up to 20 during some optimizations. For Ca$^{2+}$ to RBC velocity optimization the presence of the time shift granted a better fit of the onset phase of the responses. For Ca$^{2+}$ to fUS optimization, we used a 0–0.5 s time shift.

Transfer functions were computed between two time series (From and To). Both were cut between 5 and 27 s (for 30 s acquisitions) and between 5 and 59 s (for 60 s acquisitions). From signals were Ca$^{2+}$ responses and were interpolated at 50 ms (cubic interpolation, 'interp1' function, 'pchip' option, Matlab), as Ca$^{2+}$ data can be noisy. To signals were either RBC velocity responses, or 1 and 6 voxels Power Doppler responses. The residual sum was computed as the difference between the To signal and the convolution of the From signal and the TF, down sampled to match To.

Simulated annealing being a nondeterministic algorithm, the initial values of the parameters are crucial for the outcome and the same set of initial values can bring to different outcomes at each optimization run. We used an iterative process for which we started with an initial set of values (the ones of the first gamma-component of the standard HRF (6; 1; 0.001; 1) with the time-shift initially set to zero (0.001 for computational requirements), and run the optimization process multiple times (50–100 times). We then scored the final sets of values based on the prediction quality and we took the set with the best prediction, provided its corresponding function started at the origin of the axes and was derivable at least twice (after the time-shift time). This set was then used as an initial set for another batch of multiple runs, and so on, until the algorithm did not improve anymore the prediction made by the set of initial values in a significant manner. Usually this took no more than three iterations (see Supplementary Fig. 1).

The best set of parameter values determined the TF optimized for a given mouse in response to 1% ET, 5 s. We computed the best parameters set for all the 15 recorded mice and checked each TF to find the best one across all mice by means of a leave-one-out cross-validation. In detail, each TF, optimized in a given mouse dataset, was then used to predict the vascular responses of the rest of dataset made other mice recordings. The best TFs, finally called μTF or mTF, were chosen being the best performing on its dataset and on dataset from other mice, on the basis of the highest average Pearson coefficient and smallest coefficient of

variability, the first preferred when the two conditions were not met by any mouse. Numerical values of the optimized TF parameters are provided in Table 1.

**Analysis of data from Rungta et al.**. Vascular responses are those described as juxta-synaptic capillary in Fig. 7 of the original paper[5]. TF was computed as before, except for the fact that time series were only 15 s long. Data were collected in four OB glomeruli from three different mice[5].

**Analysis of data from Winder et al.**. Data from the original paper[27] are available on the following link: https://psu.app.box.com/v/Winder2017-Code-Data. We used the data (gamma-band LFP and IOS reflection changes in response to contralateral whisker stimulations) from the 12 mice (*_EVENTDATA_ files). We averaged every trace for both gamma-band LFP and IOS reflection changes. Data were changed into relative values and then averaged for each mouse. TF was computed as before[27].

**Prediction computation**. Micro and mesoscopic vascular predictions were computed by convoluting the Ca$^{2+}$ responses or the step functions with the correspondent TFs. An optimization of the amplitude of the responses was performed to best match predictions over experimental data. We optimized a scaling factor with a classical derivative-free method ('fminsearch' function, Matlab) using the residuals between the prediction and vascular responses from 2 to 8 s after the odor application onset as a cost function. The final prediction was then multiplied by the optimized scaling factor. Note, this post processing does not affect the Pearson coefficient we used for all the quantifications of the study.

**Custom made scripts to compute the TFs**. A simple graphical user interface version of our scripts is available on GitLab (https://gitlab.com/AliK_A/buildtf), as well as some data examples in order to give the opportunity to the reader to compute the TFs.

**Statistical tests**. All statistical analyses are reported in the manuscript, with $p$ values. A confidence level of 5% was chosen, a priori, as threshold for significant difference between samples, however higher confidence levels are reported in some figures, with graphical indicators of $p$ values.

**Reporting summary**. Further information on research design is available in the Nature Research Reporting Summary linked to this article.

## Data availability
The datasets generated and analyzed during the current study are available from the corresponding authors on reasonable request. A subset of our data is available on Zenodo (https://doi.org/10.5281/zenodo.3773863) to test our software. Source data are provided with this paper.

## Code availability
The whole code used to analyze the datasets and get the results and figures is available from the corresponding authors on reasonable request. As described in the "Methods" section, a simple version of our scripts, with a graphical user interface, and a subset of our data is available on GitLab (https://gitlab.com/AliK_A/buildtf) and on Zenodo (https://doi.org/10.5281/zenodo.3773863).

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

## Acknowledgements

Financial support was provided by the Institut National de la Santé et de la Recherche Médicale (INSERM), the European Research Council (ERC-2013-AD6; 339513), the Agence Nationale de la Recherche (ANR/NSF 15-NEUC-0003-02 and NR-16-RHUS-0004 [RHU TRT_cSVD]), the Fondation Leducq Transatlantic Networks of Excellence program (16CVD05, Understanding the role of the perivascular space in cerebral small vessel disease) and the IHU FOReSIGHT [ANR-18-IAHU-0001] supported by French state funds managed by the Agence Nationale de la Recherche within the Investissements d'Avenir program. R.L.R. had a postdoctoral fellowship award from EMBO (ALTF 384-2015). We thank Manon Omnes for mice surgeries and Christophe Tourain for technical help.

## Author contributions

A.-K.A., P.J.D., S.C., and D.B. designed the experiments. A.-K.A., W.D.H., C.P., P.J.D., S.C., and D.B. designed the analyses. C.D. and M.T. participated to the analysis of the fUS experiments. R.L.R. and P.J.D. provided the data from awake mice. A.-K.A. and D.B. wrote the customized Matlab and LabView scripts for the analyses. A.-K.A. and D.B. conducted the animal experiments and analyzed the data. Y.G.H. designed and built the co-registration setup. A.-K.A., Y.G.H., S.C., and D.B. wrote the original drafts and all authors edited the paper. S.C. and D.B. conceived the project.

## Competing Interests

M.T. is co-founder and shareholder in the ICONEUS company. The remaining authors declare no competing interests.
