## [Peer Review File · Nature Communications]

Reviewers' comments:

Reviewer #1 (Remarks to the Author):

Review of Nature Communications manuscript NCOMMS-19-1483920

"Transfer functions linking neural calcium to single voxel functional ultrasound signal"

This is follow-up study from the previously published "Mesoscopic and microscopic imaging of sensory responses in the same animal" from Drs. Charpak and Boido. This research group one of the foremost in technical imaging of the neurovascular response and in the present study played to these strengths in using optical imaging data to model the neurovascular response from neuronal activity. I agree with the authors that the underlying cascade leading to the hyperaemic response is uniquely complex and they should be commended in attempting such a study. Mathematical modelling of this system has to potential to uncover key unknowns and will be useful to the neurovascular community as a whole. Therefore, I think this work is potentially suitable for publication in Nature Communications. However, I have a few concerns/comments I would like to see the author's address before I would be able to recommend publication.

Major- scientific

This study attempts to build a model that connects changes in $[Ca^{++}]_i$ with blood flow changes as proof-of-concept that similar transfer functions can be used to model the neurovascular response more generally. Some choices in experimental design/model raise concerns with the validity of the model especially when generalising to other brain areas, tissues and experimental paradigms.

1) The authors do not provide any physiological monitoring data from the experimental animals under sedation. Blood/brain PO_2 , pH and PCO_2 have profound effects on expression of the neurovascular response (discussed PMID: 30481531). Can the authors demonstrate that they are operating within the normal ranges for these factors? If not, the applicability of the model to other experimental situations could be questioned.

Related: animals were breathing 40% O_2 during the experiment- this could mean a hyperoxic brain. Were oxygen saturation and/or blood gases monitored at all? Have the authors considered how this would change the NVR? Eg. PMID: 27499087.

2) The olfactory bulb is a highly specialised part of the brain. There is likely to be considerable heterogeneity in the NVR in throughout the brain (PMID: 21982928). The authors don't discuss how this would affect the applicability of transfer functions to other regions/tissues. Ideally, I would like to see the transfer function determined in a separate brain area with an alternative defined sensory input eg. whisker barrel where there is evidence on non-linearity in low (and high) intensity of stimulation (PMID: 15627599).

3) It was somewhat unclear how the data was divided into training and validation datasets, as is common with ML studies. Am I correct in assuming that the authors determined TFs using the entirety of the data then used model to then back-validated on the same dataset? If so, how did the authors control for the possibility of overfitting? Was the data from Boido et al, 2019 specifically excluded from the training dataset?

4) I was concerned that (in Figure 1) the TF from mouse number 1 gives the best prediction across mice. The reason for this is unclear and generally the more data (and diversity of data) in a training data set, the better the model. Why was the data not pooled? I think this needs more explanation given that the authors use this one mouse to generate the ('standard') TF used in all subsequent analysis and conclusions.

At 6% concentration of ET authors report a delayed, secondary vascular response. The genesis of this recorded vascular response may not result from an olfactory input. As it only occurs at high odour concentrations, have the authors excluded the possibility that this is activating alternative systems? For example, irritants introduced in airways can produce profound autonomic activation and subsequent changes in blood pressure may cause non-specific blood flow changes. Traces from the thermocouple output would show any changes in respiratory rate/depth that could indicate non-specific effects of odorant. Better yet recordings from outside the olfactory bulb

Did the authors try the reverse? If the blood flow response can be predicted from the $[Ca^{++}]_i$ dynamics then can the blood flow dynamics recorded be predictive of the neuronal activity? It seems that the authors have a unique opportunity to add to the literature on a much debated subject – are bloodflow changes indicative of neuronal activity?

Minor- discussion

The authors suggest (Ln 234) that TFs can be used to "...assess NVC during aging in normal and pathological animal models". Is this not a stretch here? The study shows that TFs can predict, within a single-laboratory dataset, that $[Ca^{++}]_i$ dynamics are predictive of bloodflow changes. With the important factor of amplitude being excluded, how can this claim be made for such complex phenotypes like aging without evidence?

Minor- figures

Supplementary figure 1, panel a. Authors have provided no explanation of the y-axis scale in the legend. The colour (z) scale is also incorrect.

Panel f and g also lack y-axis labels.

Minor- results

Ln95- Is simulation correct here?

Patrick Hosford, London, January 2020.

Reviewer #2 (Remarks to the Author):

The manuscript by Aydin et al describes a very careful study connecting calcium signals in neurons to the hemodynamic signals observed via functional ultrasound imaging. fUS is a very exciting new technology for non-invasive whole-brain imaging that is getting adoption in the neuroscience community. Gaining a better understanding of how to interpret fUS signals in terms of underlying neural activity is therefore broadly interesting and valuable. This paper does a nice job of achieving this goal. In particular, I like the way they develop transfer functions (TF) linking GCaMP signal changes to both microscale (observed optically) and macroscale (fUS) hemodynamics, and test these connection across a range of stimuli strengths and durations in the olfactory bulb. In addition to characterizing the temporal relationship, they also look at how precisely fUS can discern the spatial pattern of activation. Overall, this is an excellent study that will interest the readers of NComms.

I just have a few minor suggestions for the authors to consider:

Since in Fig 4a, you have both microscale and fUS hemodynamic responses measured together,

why not also create a TF linking the two? Right now, readers can visually compare the μ TF to the MTF in Fig. 4b, but it might be nice for them to also see the relationship between the two more directly.

Is the algorithm described for finding TFs really “machine learning”. This is not my area, but it seems more like a conventional parameter-fitting approach. I just ask the authors to consider whether they really mean ML or are using this term gratuitously to “spice up” the paper.

Line 153: system → systems

Line 167 says the mTF had a slower decay than the μ TF, but Fig 4b shows the opposite. Which is correct?

It would be helpful to readers to clarify in the legend and labeling for Fig 4c that what is being plotted is Prediction Quality.

It would be helpful if the authors could comment on why other parts of the image in Fig 3c — far away from the main glomerulus and its surrounding 6 pixels — are being activated just as strongly, and whether there are any insights on how this activation is temporally related to the glomerulus calcium signal.

Fig 3b: I would add x, y, z, labels to the depicted axes (which are very small to see and recognize from Fig 3a).

Serge Charpak,
Vision Institute, Sorbonne University
17 rue Moreau, Paris 75012
Telephone: +33 6 62 45 05 89
Email: serge.charpak@inserm.fr

Paris, March 30th, 2020

Responses to Reviewer #1:

This is follow-up study from the previously published “Mesoscopic and microscopic imaging of sensory responses in the same animal” from Drs. Charpak and Boido. This research group one of the foremost in technical imaging of the neurovascular response and in the present study played to these strengths in using optical imaging data to model the neurovascular response from neuronal activity. I agree with the authors that the underlying cascade leading to the hyperaemic response is uniquely complex and they should be commended in attempting such a study. Mathematical modelling of this system has to potential to uncover key unknowns and will be useful to the neurovascular community as a whole. Therefore, I think this work is potentially suitable for publication in Nature Communications. However, I have a few concerns/comments I would like to see the author’s address before I would be able to recommend publication.

We thank Reviewer #1 for his positive general comments and his comments

This study attempts to build a model that connects changes in $[Ca^{++}]_i$ with blood flow changes as proof-of-concept that similar transfer functions can be used to model the neurovascular response more generally. Some choices in experimental design/model raise concerns with the validity of the model especially when generalizing to other brain areas, tissues and experimental paradigms.

We agree with reviewer #1 that our work, a priori, cannot be generalized to other brain regions, brain states and other species, without demonstrating it. It was not our goal as we believe that it is already crucial to control perfectly a single experimental condition in order to support our claims. However, we understand the need to extend our findings and we are now demonstrating the extent to which our TF approach can be applied to different experimental conditions and brain regions (see below).

1) The authors do not provide any physiological monitoring data from the experimental animals under sedation. Blood/brain PO₂, pH and PCO₂ have profound effects on expression of the neurovascular response (discussed PMID: 30481531). Can the authors demonstrate that they are

operating within the normal ranges for these factors? If not, the applicability of the model to other experimental situations could be questioned.

Related: animals were breathing 40% O₂ during the experiment- this could mean a hyperoxic brain. Were oxygen saturation and/or blood gases monitored at all? Have the authors considered how this would change the NVR? Eg. PMID: 27499087.

Reviewer #1 is right in mentioning that resting homeostasis is important when studying neurovascular coupling and that 40% O₂ is higher than what humans and animals normally breathe (21 % O₂). On the other hand, these experiments were performed under anesthesia which is known to decrease blood PO₂ (beside light isoflurane anesthesia which increases blood PO₂, Lyons et al, 2016). In our hands, inhalation of 40 % O₂ (under anesthesia) does not affect neurovascular coupling, as previously shown by several papers, in particular the paper by Lindauer et al (JCBFM 2010, doi: 10.1038) which clearly showed that neurovascular coupling occurs in a very similar manner at extremely high PO₂ (hyperbaric hyperoxygenation up to 4 standard atmospheres of absolute pressure). So we believe that resting PO₂, in capillaries or in the neuropil, has very little influence, if any, on neurovascular coupling. Concerning the role of activity-evoked drops of PO₂ on neurovascular coupling (Wei et al. PMID: 27499087), we find that this hypothesis is not supported by our data in the olfactory bulb. First, even though PO₂ rapidly drops in activated glomeruli (doi: 10.1523/JNEUROSCI.4817-08.2009, doi: 10.1038/nm.3059) by about 2.5 mmHg/s, upon very high odor concentration, this O₂ dip rarely exceeds 2-3 mm Hg (Lecoq et al 2009 and Parpaleix et al 2013). On the other hand, PO₂ inter RBC, which reflects juxta-capillary tissue PO₂, constantly fluctuates with a SD of 2.6 mmHg (Lyons et al. 2016). Finally, we recently reported that there is no threshold of neuronal activation (very low odor stimulation) below which neurovascular coupling does not occur (Boido et al. DOI: [10.1038/s41467-019-09082-4](https://doi.org/10.1038/s41467-019-09082-4)). At these odor concentrations, PO₂ drops are not measurable making unlikely a major role of odor evoked PO₂ drop in NVC.

To reassure reviewer #1 that our conditions are physiological, we have added the analysis of data obtained in the olfactory bulb of awake trained animals. These data have already been used for another purpose in Rungta et al (2018) and it is clearly indicated as such in the manuscript (line 138). As it can be seen in **Supplementary Fig.4a**), the quality of the prediction remains high (Pearson coefficients of 0.91, 0.84, 0.77, 0.92). Although in 2 cases over 4, vascular responses displayed a delayed negativity that was never observed in anesthetized animals, the positive component is well-fitted. This stresses that anesthesia, as expected, modulates NVC (see also below data from the cortex).

2) The olfactory bulb is a highly specialized part of the brain. There is likely to be considerable heterogeneity in the NVR in throughout the brain (PMID: 21982928). The authors don't discuss how this would affect the applicability of transfer functions to other regions/tissues. Ideally, I would like to see the transfer function determined in a separate brain area with an alternative defined sensory input eg. whisker barrel where there is evidence on non-linearity in low (and high) intensity of stimulation (PMID: 15627599).

The olfactory bulb is a specialized brain region in that there is a huge convergence of glutamatergic terminals per glomerulus, and odors generate large local vascular responses. We don't believe that the mechanisms underlying neurovascular coupling are specific to the olfactory bulb. Nevertheless, we certainly agree that our TFs cannot be readily applied to other brain regions, without demonstrating it. To address the issue of the generalization of our approach to the cortex, we provide

new analysis of intrinsic imaging responses (total hemoglobin) to whisker stimulation, in the awake mouse. As above, the data have also been previously used (Winder et al. 2017), but we applied our transfer function approach. **Supplementary Fig 4b** shows that even though 1) the sensory modality differed, 2) calcium signals were substituted with changes of neuronal activity in the gamma band, 3) CBV was estimated from the intrinsic optical signal (IOS) reflection changes ($\Delta R/R$), our TF predictions remained astonishingly good (Pearson coefficients > 0.7). In many cases, IOS signals showed a delayed undershoot (as in the olfactory bulb of awake mice), which could not be fitted with a single gamma function. However, the optimization of new TFs significantly improved the predictions. These results have been added to the manuscript (lines 181-192). To conclude, our approach of TF optimization is efficient to compute TFs across brain state, brain regions and with various inputs/outputs. Moreover, the TFs *per se*, optimized from the olfactory bulb in anesthetized mice remain solid in the awake OB and cortex, even when changing the input and output signal. Finally, we agree that it would be very interesting to test for the linearity of whisker barrel responses by recording calcium and RBC velocity responses at several stimulation intensities. However, it would have taken an additional full year of work, and we have had to kill all our transgenic animals, trained or not, due to the confinement imposed by the Covid-19 pandemic.

3) It was somewhat unclear how the data was divided into training and validation datasets, as is common with ML studies. Am I correct in assuming that the authors determined TFs using the entirety of the data then used model to then back-validated on the same dataset? If so, how did the authors control for the possibility of overfitting? Was the data from Boido et al, 2019 specifically excluded from the training dataset?

We thank the reviewer for pointing the lack of information for the general reader. This is now better formulated in first subsection of the results. To cross-validate our observations, we adopted the leave-one-out cross-validation (LOOCV) approach that is a common choice for a 15 mice data set, an acceptable size for an *in vivo* biological assay but a relatively small value for machine-learning algorithms. With the LOOCV, each mouse is a ‘training’ set by itself, that is compared to other mice data as a test set. The ‘best’ TF was the one performing the better overall (as we explained in the Methods section).

As such, we did not train and validate our model on the same datasets. Moreover, overfitting is usually due to a high degree of freedom in the model relative to the number of parameters. We avoid this issue by fitting a single gamma function, which has at most 4 free parameters, and we are using 500-1000 data points per optimization. Overall, our approach makes extremely unlikely any overfitting issue and allows us to conclude on the robustness of NVC.

Regarding the data from Boido et al., we found TFs performing very well in all the ‘original’ tested mice. Since the experimental paradigm was the same, we decided to add these ‘old’ data to keep increasing the validation of our model. These mice are now clearly indicated in the panel (mice # 12 to 15)

4) I was concerned that (in Figure 1) the TF from mouse number 1 gives the best prediction across mice. The reason for this is unclear and generally the more data (and diversity of data) in a training data set, the better the model. Why was the data not pooled? I think this needs more explanation given that the authors use this one mouse to generate the (‘standard’) TF used in all subsequent analysis and conclusions.

Reviewer #1 is right: ‘the more data (and diversity of data) in training data set, the better the model’. In fact, as we explained above, we used every single mouse for training and we tested it towards all the other mice, sorting out the ‘best’ one this way. This is the principle of the LOOCV approach. As we now better explain in the methods section, “The best TFs, finally called μ TF or MTF, were chosen, being the best performing on its data-set and on data-set from other mice, on the basis of the highest average Pearson coefficient and smallest Coefficient of Variability, the first preferred when the two conditions were not met by any mouse”. Note that we averaged only individual trials of calcium responses for a given odor stimulation, and used this average response to predict the corresponding averaged RBC velocity response in the same animal (**Fig. 1e**) With this approach, we find that whatever the heterogeneity of calcium and RBC velocity response dynamics in all the mice, predictions remain excellent. Because there is variability in the hemodynamic responses (some slower and some faster) pooling all data would produce ‘hybrid’ dynamics, without real physiological meaning. Other cross-validation options, like the leave-p-out, would require a high computational time which, given the high similarity of the computed TFs (at both micro and mesoscopic levels), would not bring any significative advantage in the search for the best TF.

At 6% concentration of ET authors report a delayed, secondary vascular response. The genesis of this recorded vascular response may not result from an olfactory input. As it only occurs at high odour concentrations, have the authors excluded the possibility that this is activating alternative systems? For example, irritants introduced in airways can produce profound autonomic activation and subsequent changes in blood pressure may cause non-specific blood flow changes. Traces from the thermocouple output would show any changes in respiratory rate/depth that could indicate non-specific effects of odorant. Better yet recordings from outside the olfactory bulb.

Reviewer #1 points to a concern we constantly have. The best way to discard an activation of “alternative systems” is indeed to analyze our monitoring of respiration frequency. We have gone through our experiments where high odor concentration triggers delayed vascular responses and we have quantified the instantaneous respiration frequency during odor (see **Figure 1** below, not included in the MS). As it can be observed, we have no indication that irritation or any side effect affects respiration. Our data thus support the hypothesis that strong stimulation may trigger a secondary biphasic response involving astrocytes, as initially reported by Shulz et al. (2012) and Tran et al. (2018). We did not include the figure in the manuscript but mentioned the point in the discussion (line 248).

Did the authors try the reverse? If the blood flow response can be predicted from the $[Ca^{++}]_i$ dynamics then can the blood flow dynamics recorded be predictive of the neuronal activity? It seems that the authors have a unique opportunity to add to the literature on a much debated subject – are blood flow changes indicative of neuronal activity?

Having the TF and blood flow responses, we can indeed deconvolve calcium responses from vascular responses. However, we are very cautious about applying the reverse approach for one main reason, which in fact concerns the whole brain vascular arbor: due to vascular backpropagation along endothelial cells: the brain volume irrigated by functional hyperemia is larger than that of activated neurons. There is thus a risk to ascribe a “false” neuronal activation to brain voxels showing a vascular change.

Minor- discussion

The authors suggest (ln 234) that TFs can be used to “...assess NVC during aging in normal and pathological animal models”. Is this not a stretch here? The study shows that TFs can predict, within a single-laboratory dataset, that $[Ca^{2+}]_i$ dynamics are predictive of blood flow changes. With the important factor of amplitude being excluded, how can this claim be made for such complex phenotypes like aging without evidence?

We initially estimated that we could raise such hypothesis. As we concede that we did not provide any data supporting it, we have removed this prospective remark.

Minor- figures

Supplementary figure 1, panel a. Authors have provided no explanation of the y-axis scale in the legend. The colour (z) scale is also incorrect.

Done

Panel f and g also lack y-axis labels.

Done

Minor- results

Ln95- Is simulation correct here?

Now it has been corrected: stimulation.

Responses to Reviewer #2:

The manuscript by Aydin et al describes a very careful study connecting calcium signals in neurons to the hemodynamic signals observed via functional ultrasound imaging. fUS is a very exciting new technology for non-invasive whole-brain imaging that is getting adoption in the neuroscience community. Gaining a better understanding of how to interpret fUS signals in terms of underlying neural activity is therefore broadly interesting and valuable. This paper does a nice job of achieving this goal. In particular, I like they way they develop transfer functions (TF) linking GCaMP signal changes to both microscale (observed optically) and macroscale (fUS) hemodynamics, and test these connection across a range of stimuli strengths and durations in the olfactory bulb. In addition to characterizing the temporal relationship, they also look at how précised fUS can discern the spatial pattern of activation. Overall, this is an excellent study that will interest the readers of NComms.

We thank Reviewer #2 for his/her very kind general comments.

I just have a few minor suggestions for the authors to consider:

Since in Fig 4a, you have both microscale and fUS hemodynamic responses measured together, why not also create a TF linking the two? Right now, readers can visually compare the μ TF to the MTF in Fig. 4b, but it might be nice for them to also see the relationship between the two more directly.

Such transfer function would link microscopic vascular responses in glomerular capillaries to fUS single/6 voxel responses. We have attempted to create this TF (**Figure 2** below, not included in the MS). As it can be seen, RBC and fUS responses are so similar in time that a Dirac Delta function can be used to link the two types of responses. In other words, the main possible difference between RBC velocity and fUS signals is, eventually, a very short time-shift, below our temporal resolution. For this reason, we mention the point in the discussion but did not add the figure.

Is the algorithm described for finding TFs really “machine learning”. This is not my area, but it seems more like a conventional parameter-fitting approach. I just ask the authors to consider whether they really mean ML or are using this term gratuitously to “spice up” the paper.

Due to the nature of the function used for the fit (single-gamma driven HRF), as we showed in the supplementary figure 1a, the parameters act on the shape of the function in a concerted manner: for instance 3 out of 4 parameters modify, directly or not, the amplitude. This brings to an ill-posed optimization problem with several local minima. Deterministic optimization algorithms would fail in correctly fitting the function, depending on the chosen initial values for the parameters. With our first approach (quasi-Newton algorithm), we often fell into a TF with a zero onset or peak (**Fig. 3 upper panel**, below) or in local minima not as deep as the ones reachable with non-deterministic algorithms (**Fig. 3 lower panel**).

A non-deterministic approach, like simulated annealing, frequently used in machine learning algorithms, allowed us to avoid most of the local minima. Then, the iterative optimization and selection of the TF, together with the leave-one-out cross-validation, are further ‘ingredients’ (Patrick, ingredient?) of a common machine learning approach. We used the definition of machine-learning to describe in a compact manner the type of analytical method we adopted and its novelty with respect to similar studies, without any ‘fashion’ purpose. We agree that, with different biological signals or choosing other functions for the fit, or if we had accepted to smooth more our signals, we could have used classical fitting methods, as used in other papers (Sirotin & Das, Nature, 2009), but we opted to stick on the single-gamma HRF because it is a ‘standard’ choice to model NVC and it is already implemented in most software for neuroimaging data analysis.

Line 153: system —> systems

Corrected

Line 167 says the mTF had a slower decay than the μ TF, but Fig 4b shows the opposite. Which is correct?

Indeed, we wrote that the m TF has a slower decay, which is what is shown in fig.4b as the decay is longer. We believe reviewer #2 had a misunderstanding here.

It would be helpful to readers to clarify in the legend and labeling for Fig 4c that what is being plotted is Prediction Quality.

We have added this notion of quality in the legend of **Fig. 4**.

It would be helpful if the authors could comment on why other parts of the image in Fig 3c — far away from the main glomerulus and its surrounding 6 pixels — are being activated just as strongly, and whether there are any insights on how this activation is temporally related to the glomerulus calcium signal.

Reviewer #2 points to a specificity of the olfactory bulb. In the neocortex where a brief stimulation of a single whisker generates a local neuronal activation that spreads over the entire barrel cortex within few tens of milliseconds (Ferezou et al, • DOI: [10.1016/j.neuron.2006.03.043](https://doi.org/10.1016/j.neuron.2006.03.043)). This involves cortical poly-synaptic activation. In the olfactory bulb, an additional mechanism is responsible of the spatial activation spread: odorant receptors are not so specific when the odor concentration is far from threshold. Consequently, at the odor concentration used in our work (ET 1%), we certainly activated several olfactory receptor neuron populations and thus activated many glomeruli in the olfactory bulb coronal slice. As the “non-specific” activation of olfactory neurons in the nasal cavity is rapid, the temporal activation of voxels, others than the 6 voxels centered on the most sensitive glomerulus is very similar. This is in line with what we have previously reported (see **Fig 5** in Boido et al., 2019) and is also shown in **Figure 4** below (not included in the MS to avoid redundancy with the previous paper) where we show responses of 3 ROI that did not comprise the previous single/6 voxels. Clearly, at ET 1%, a large part of the dorsal olfactory bulb responds to odor. The point is mentioned in the discussion lines 276-277.

Fig 3b: I would add x, y, z, labels to the depicted axes (which are very small to see and recognize from Fig 3a).

Done

Best regards,

Serge Charpak

Figure 1. High odor concentration does not induce changes in breathing in the anesthetized mouse.

(a) A thermocouple recording from a typical single acquisition. Orange dots are the peaks detected by Matlab *findpeaks* function to compute the frequency (only the first 25 seconds of acquisition are shown). ET was applied at high concentration (6 %) during 5 s (grey shade), a condition that systematically generates a second vascular component. (b) Instantaneous frequency computed from the recording in (a). The odor inhalation is devoid of any effect. (c) Mean instantaneous frequency (average of 3-5 trials) over time for the 5 mice used in Figure 4. Breathing frequency is stable over the acquisition.

Figure 2. A simple Dirac delta function links RBC velocity and fUS responses. Using the data from Fig. 4a, our TF algorithm optimized a simple Dirac which, convolved with RBC responses (light blue traces) predicted (red traces) almost perfectly fUS responses (dark blue responses) The Dirac having a 0 delay, there is only a scale factor between the two vascular responses.

Figure 3. Comparison between two different algorithms to compute TF.
(Upper panel) Optimization made with a quasi-newton algorithm (*fmincon*, Matlab). The computed TF shows a non-derivable peak which is not compatible with a biological mechanism.
(Lower panel) Optimization made with a non-deterministic approach (*simulannealbnd*, Matlab). It allows to reach for the global minimum and not local ones.

Figure 4. ET 1 % 5 s activates a major part of the dorsal olfactory bulb.
(a) Activation map from **Fig.3c**. Three ROIs were chosen apart from the 6 voxels centered on the most responsive glomerulus. **(b)** The three ROIs showed CBV responses to odor stimulation (orange background).

REVIEWERS' COMMENTS:

Reviewer #1 (Remarks to the Author):

I wish to thank the authors for addressing my comments and suggestions so thoroughly. They have gone to great lengths to include extra analysis from their own and external data sets to allay my concerns regarding the generalizability of their model. Additionally, by inclusion of recordings in awake animals (as well as additional controls) they have shown that the model is applicable irrespective of aesthetic state or underlying basal physiology. I can now recommend this paper for publication.

The loss of the author's animal colony is indeed lamentable and I wish them speedy recovery from the present disruption.